# The Mulberry *SPL* Gene Family and the Response of *MnSPL7* to Silkworm Herbivory through Activating the Transcription of *MnTT2L2* in the Catechin Biosynthesis Pathway

**DOI:** 10.3390/ijms23031141

**Published:** 2022-01-20

**Authors:** Hongshun Li, Bi Ma, Yiwei Luo, Wuqi Wei, Jianglian Yuan, Changxin Zhai, Ningjia He

**Affiliations:** State Key Laboratory of Silkworm Genome Biology, Southwest University, Beibei, Chongqing 400715, China; hong_shun_li@163.com (H.L.); mbzls@swu.edu.cn (B.M.); luoyiwei12@swu.edu.cn (Y.L.); swuwuqi@email.swu.edu.cn (W.W.); yuanjiangl@swu.edu.cn (J.Y.); zhaichangxin1996@163.com (C.Z.)

**Keywords:** mulberry, *SPL*s, phylogenetic analysis, silkworm herbivory, *MnSPL7*/*MnTT2L2* module

## Abstract

*SQUAMOSA PROMOTER BINDING PROTEIN-LIKE* (*SPL*) genes, as unique plant transcription factors, play important roles in plant developmental regulation and stress response adaptation. Although mulberry is a commercially valuable tree species, there have been few systematic studies on *SPL* genes. In this work, we identified 15 full-length *SPL* genes in the mulberry genome, which were distributed on 4 *Morus notabilis* chromosomes. Phylogenetic analysis clustered the *SPL* genes from five plants (*Malus × domestica Borkh*, *Populus trichocarpa*, *M. notabilis*, *Arabidopsis thaliana*, and *Oryza sativa*) into five groups. Two zinc fingers (Zn1 and Zn2) were found in the conserved SBP domain in all of the *MnSPL*s. Comparative analyses of gene structures and conserved motifs revealed the conservation of *MnSPLs* within a group, whereas there were significant structure differences among groups. Gene quantitative analysis showed that the expression of *MnSPLs* had tissue specificity, and *MnSPLs* had much higher expression levels in older mulberry leaves. Furthermore, transcriptome data showed that the expression levels of *MnSPL7* and *MnSPL14* were significantly increased under silkworm herbivory. Molecular experiments revealed that *MnSPL7* responded to herbivory treatment through promoting the transcription of *MnTT2L2* and further upregulating the expression levels of catechin synthesis genes (*F3′H*, *DFR*, and *LAR*).

## 1. Introduction

*SQUAMOSA PROMOTER BINDING PROTEIN-LIKE* (*SPL*) genes encode plant-specific transcription factors and are typified by highly conserved SQUAMOSA promoter-binding (SBP) domains. SBP domains contain 75–79 amino acid residues with two Zn^2+^-binding sites (Cys–Cys–His–Cys and Cys–Cys–Cys–His) and a nuclear location signal (NLS) which is involved in DNA binding and nuclear localization [1,2]. *SPL* genes were first identified in the floral organs of *Antirrhinum majus* while screening the nuclear protein interactions within the promoter of the *SQUAMOSA* gene by electrophoretic mobility shift assay (EMSA) [3].

Importantly, *SPL* genes are distributed in a vast majority of green plants, including single-celled algae, mosses, gymnosperms, and angiosperms [4,5,6]. To date, since the completion of genome sequencing and the thorough development of the functional genome, 17, 19, 56, 16, and 14 *SPL* genes have been systematically identified in *A. thaliana, O. sativa*, *Triticum aestivum*, *Peaonia suffruticosa,* and *Paeonia suffruticosa,* respectively [7,8,9,10,11].

As unique plant transcription factors, *SQUAMOSA PROMOTER BINDING PROTEIN-LIKE* (*SPL*) genes play important roles in plant vegetative phase transition [12,13,14], flowering regulation [15,16], leaf morphogenesis [17], and root development [18,19,20]. Moreover, *SPL* genes can respond to various biotic and abiotic stresses by regulating the abundances of anthocyanin [21,22], abscisic acid [23], and jasmonate [24]. *SPL* genes have a higher expression level in relatively more mature tissues. In *Antirrhinum, SPL1* and *SPL2* were detected in inflorescences but not in the tissues of juvenile plants [25]. In *Arabidopsis*, the expression profiles of *SPL* genes increased as plants aged and were regulated by sequence-conserved microRNAs (miR156/157) [12,15,26,27]. *AtSPL9* and *AtSPL10*, which were directly repressed by miR156/157, controlled the juvenile-to-adult phase transition [12]. *AtSPL3/4/5* were also directly suppressed by miR156/157 and positively regulated floral meristem identity and influenced the trichome distribution [16,28]. Additionally, *AtSPL9* and *AtSPL15* affected leaf shape [12], while *AtSPL3*, *AtSPL9*, and *AtSPL10* were involved in the repression of lateral root growth, and *AtSPL10* played a dominant role in the primary root meristem activity regulation [18,19,20]. In addition, the expression levels of *SPL* genes fluctuated significantly under adverse conditions including salt stress, drought stress, heat stress, nitrogen starvation, and viral defense [29,30,31]. The overexpression of miR156 repressed the expression of *SPL* genes and further increased anthocyanin accumulation in the stems of *Arabidopsis* and the stem apex of poplar. The increased expression of miR156 also improved the accumulation of flavonoids in poplar [21,22,32]. In *Arabidopsis*, high levels of *SPL9* under salt and drought stress treatment suppressed anthocyanin accumulation by directly repressing the expression of anthocyanin biosynthetic genes, such as *ANS*, *F3′H*, *DFR*, and *UGT75C1*, through interfering with the integrity of the MYB-bHLH-WD40 transcriptional activation complex [21,22].

Mulberry, also known as *Morus alba*, is widely distrusted around the world [33]. In addition to its use for breeding silkworms, mulberry is also a traditional fruit tree and Chinese herbal medicine that has significant economic value in food and medicine production [34,35,36]. Abundant flavonoids have been detected in mulberry leaves and fruits, which indicates the medicinal properties and high stress resistance of mulberry [37,38]. Research has shown that flavonoid (anthocyanins, proanthocyanidins, flavones, and flavonols) accumulation is regulated by the MYB-bHLH-WD40 ternary activation complex in plants [39]. In mulberry, the TT2L1 and TT2L2 proteins interact with bHLH3 or GL3 to promote proanthocyanidin (the multimer of catechin or epicatechin) accumulation [40]. Furthermore, mulberry genome sequencing has been completed [33,41], and our previous study also verified that nine mulberry *SPL* genes are directly regulated by miR156 [42]. However, to date, systematic research on mulberry *SPL* genes is still lacking; consequently, the specific molecular biological functions of SBP-box transcription factor genes in development and stress responses remain unclear.

In this work, we identified 15 *SPL* genes in the mulberry genome and named these genes according to their evolutionary relationship with *AtSPL* genes. The *MnSPL* gene family was characterized through comprehensive analyses of gene structures, phylogenetic relationships, chromosomal locations, conserved motifs, and expression patterns. A dual-luciferase assay was carried out to verify the interaction between the *MnSPL7* gene and catechin synthesis-regulated genes (*TT2L2*) under silkworm herbivory in mulberry. In summary, our works have provided basic information for elucidating the molecular biological functions of *SPL* genes in mulberry.

## 2. Results

### 2.1. Identification and Analysis of the SPL Gene Family in Mulberry

A total of 15 full-length *SPL* genes were identified in the mulberry genome (MorusDB, https://morus.swu.edu.cn/, accessed on 3 March 2020) [43] using BLASTAN and HMMER with *SPL* genes from *A. thaliana* as the query sequences (Table 1). Mapping *SPLs* to the *M. notabilis* genome showed that 15 *SPLs* were unevenly distributed on 4 chromosomes, with 5 on Chr1 (*SPL1*, *SPL3*, *SPL4*, *SPL8*, and *SPL10*) and Chr2 (*SPL2*, *SPL6*, *SPL7*, *SPL14*, and *SPL16B*), 3 on Chr4 (*SPL5*, *SPL15*, and *SPL16A*), and 2 on Chr6 (*SPL12* and *SPL13*) (Figure 1).

To investigate the evolutionary relationships of *SPL* genes in plants, we collected a dataset of 104 putative SPL protein sequences, including 18 from rice, 16 from *Arabidopsis*, 28 from poplar, 27 from apple, and 15 from mulberry, for a phylogenetic analysis with a neighbor-joining (NJ) phylogenetic tree. The result of phylogenetic analysis showed that these 104 *SPL* genes were relatively evenly clustered into 5 groups (G1–G5) and each group contained at last 1 SPL protein from these 5 species. However, MnSPLs had proximate relationships to MdSPLs and outermost relationships to SPLs from rice (Figure 2). These results suggest that plant SPL genes may have originated from common ancestral genes, but some SPL genes may have been differentiated separately between monocots and eudicots. Nine *MnSPL* genes have been verified as the target genes of miR156 in mulberry [42]. Here, we found that most of the miR156-target *MnSPLs* clustered with miR156-target *AtSPLs*, except for *MnSPL7* and *MnSPL14* (Figure 2).

Gene structure analysis revealed that all the *MnSPL* genes contained the SBP domain, and the number of exons in 15 *MnSPL* genes varied from 2 to 12. The number of *MnSPLs* with 2, 3, 4, 10, and 12 exons were 3 (*MnSPL3*, *MnSPL4*, and *MnSPL5*), 7 (*MnSPL6*, *MnSPL8*, *MnSPL10*, *MnSPL13*, *MnSPL15*, *MnSPL16A*, and *MnSPL16B*), 1 (*MnSPL2*), 3 (*MnSPL1*, *MnSPL12*, and *MnSPL14*), and 1 (*MnSPL7*), respectively. There were 9 miR156-target *SPL* genes in mulberry, 7 of which had miR156 recognition sites in the exon region in addition to 1 in the intron region and 1 in the 5′-UTR region (Figure 3a and Table 1). The SBP domains in MnSPLs had 75 amino acid residues. Sequence analysis of the SBP domains in MnSPLs revealed that the conserved zinc binding sites, the zinc fingers Zn1 and Zn2, also existed in the SBP domains. In addition to zinc binding sites, the SBP domains also contained a conserved NLS in the C-terminus of the SBP domains (Figure 3b). Furthermore, other conserved motifs were searched using the online tool MEME with the default parameters. The results showed that 20 conserved motifs were identified in 15 MnSPLs, and genes in the same group had highly similar motif distribution (Figure 4).

### 2.2. Temporal-Spatial Expression Profile Analysis of SPL Genes in Mulberry

To uncover the potential biological functions of *MnSPL* genes, we gathered the read data of these genes in five different mulberry tissues (roots, branch bark, winter buds, male flowers, and leaves) from the MorusDB (https://morus.swu.edu.cn/, accessed on 3 March 2020). Statistical results showed that these 15 *MnSPL* genes had tissue-preferential expression in 5 samples. Three *SPL* genes (*MnSPL1*, *MnSPL13*, and *MnSPL14*) were significantly highly expressed in roots. *MnSPL12* was expressed mainly in branch bark, *MnSPL4*, and *MnSPL5* were expressed mainly in leaf tissues, 6 genes (*MnSPL3*, *MnSPL6*, *MnSPL8*, *MnSPL10*, *MnSPL15*, and *MnSPL16A*) had significantly high expression levels in winter buds, and 3 genes (*MnSPL2*, *MnSPL7*, and *MnSPL16B*) were prominently highly expressed in male flowers (Figure 5 and Appendix A). Then, we tested the expression profiles of *MnSPL* genes in the mulberry root, bark, and leaf tissues at the juvenile and mature phases. We found that all the *MnSPL* genes had higher expression levels in mature leaf tissue than in juvenile leaf tissue, except for *MnSPL2*. In mulberry roots, five *SPL* genes (*MnSPL1*, *MnSPL4*, *MnSPL7*, *MnSPL14*, and *MnSPL16A*) showed no difference in expression between juvenile and mature samples. The expression level of three genes (*MnSPL6*, *MnSPL12*, and *MnSPL16B*) decreased with age, and two genes (*MnSPL13* and *MnSPL15*) had a higher expression level in the older root tissue. Most of the *MnSPL* genes showed no difference in expression between juvenile and mature mulberry bark, except for two genes (*MnSPL4* and *MnSPL13*) that were more highly expressed in mature bark and two genes (*MnSPL16A* and *MnSPL16B*) that had lower expression in mature bark (Figure 6). The tissue-specificity expression profile implied that *MnSPL* genes had functional diversity. The expression levels of *MnSPLs* were increased with age, which suggested that the biological functions of *MnSPLs* were regulated by age in mulberry leaves.

### 2.3. Silkworm Herbivory Influenced the Expression Profile of MnSPL Genes

Plants balance their energy assignment between development and stress responses to ensure their survival using the miR156-SPL module [24,44]. The co-evolution between mulberry and silkworm was influenced by artificial selection for thousands of years. We were curious to learn whether mulberry *SPLs* responded to silkworm herbivory. Transcriptome data showed that the expression levels of *SPL7* and *SPL14* significantly increased, and five *SPL* genes (*SPL2*, *SPL12*, *SPL16A*, *SPL5*, and *SPL15*) prominently decreased under herbivory in wild mulberry leaves (Chuansang, *M. notabilis*) (Figure 7c and Appendix A). The results of RT-qPCR showed that except for *SPL12* and *SPL14*, all *SPL* genes were significantly less expressed under herbivory treatment in cultivated mulberry leaves (Guisangyou 62, *M. atropurpurea cv. Guisangyou 62*) (Figure 7d and Appendix A). Further quantitative analysis showed that the expression levels of miR156 in both cultivated and wild mulberry leaves significantly increased after herbivory treatment (Figure 7a,b). These results indicated that the miR156-SPL module responded to silkworm herbivory in both cultivated and wild mulberry; however, *SPL7* and *SPL14* were more highly expressed, independent of the miR156-SPL module, in the wild mulberry leaves under herbivory.

In identify the molecular mechanism of *SPL* in mulberry leaves under silkworm herbivory treatment, we analyzed the transcriptome data of wild type and herbivory treatment leaves from Chuansang. The results revealed that genes (*TT2L2*, *bHLH*, *TTG1*, *F3′H*, *DFR*, and *LAR*) associated with catechin (the monocase of procyanidine) synthesis were significantly more highly expressed in leaves under herbivory (Figure 8a,b). However, the results of RT-qPCR showed that the expression levels of *F3′H*, *DFR*, and *LAR* in Guisangyou 62 leaves decreased under herbivory treatment (Figure 8c). Promoter analysis of genes (*TT2L2*, *bHLH*, *TTG1*, *F3′H*, *DFR*, and *LAR*) associated with catechin synthesis revealed that there was a predicted ggaCGTACa *cis-acting* element on the promoter of the *TT2L2* gene in *M. notabilis*, which could be recognized by the SBP domain of *SPL* genes (data not shown). The dual-luciferase assay verified that SPL7 could combine with the promoter of *TT2L2* and promoted its transcription (Figure 9). Taken together, these results suggested that SPL7 responded to herbivory treatment through promoting the transcription of *TT2L2* in wild mulberry (Chuansang), and this interaction was not detected in cultivated mulberry (Guisangyou 62).

## 3. Discussion

### 3.1. The Evolutionary Conservation and Functional Diversity of SPL Genes in Mulberry

Based on phylogenetic analysis, 107 SPL proteins from *Malus × domestica Borkh*, *P. trichocarpa*, *M. notabilis*, *A. thaliana,* and *O. sativa* were clustered into 5 groups. Each group contained at least one sequence from all five species, even though there were visible differences in gene number among these five plants, which indicated that the ancestral gene of *SPLs* already existed before the speciation between monocot and dicot plants (Figure 2). Moreover, the NJ phylogenetic tree showed that most mulberry SPLs were classed together with MdSPLs, and mulberry SPLs had the farthest genetic distance to OsSPLs (Figure 2). *SPL* genes encoded SPL proteins with a highly conserved DNA-binding domain named the SBP domain [1]. Systematic comparative analysis of mulberry *SPL* genes revealed that the highly conserved SBP domain had about 75 amino acid residues (Figure 3 and Figure 4), which indicated the evolutionary conservation of mulberry *SPL* genes.

In *Arabidopsis*, *SPL2*, *SPL9*, *SPL10*, *SPL11*, *SPL13* and *SPL15* exhibit functional redundancy in both the vegetative phase transition and the vegetative-to-reproductive transition [26]. This study found that mulberry SPL2, SPL10, and SPL15 were clustered into group 5 with AtSPL2, AtSPL9, AtSPL10, AtSPL11 and AtSPL15 in the phylogenetic tree (Figure 2) and were significantly expressed in the reproductive organs (winter buds and male flowers) in mulberry (Figure 5). Based on the above data, we inferred that the *SPL2*, *SPL10* and *SPL15* genes also contributed to reproductive growth in mulberry. The tissue differential expression profiles of mulberry *SPL* genes among winter buds, male flowers, roots, branch bark, and leaves were also detected in this work, just as *PtSPLs* [7], *AtSPLs* [26], and *OsSPLs* [45], which implied the functional diversity of *SPL* genes in mulberry (Figure 5). The expression levels of *AtSPL* genes increased with Arabidopsis aging [12,15]. Similarly, we determined that *SPL* genes had a higher expression level in the older mulberry leaves (Figure 6a), which indicated that the function of mulberry *SPL* genes was also influenced by age.

### 3.2. The MnSPL7/MnTT2L2 Module Responds to Silkworm Herbivory through Regulating Catechin Synthesis Gene Expression in Wild Mulberry (Chuansang)

In *Arabidopsis*, *AtSPL9* took part in responses to drought and salt stress by influencing anthocyanin metabolism [22]. We also found that the expression levels of mulberry *SPL* genes fluctuated after silkworm herbivory treatment. *SPL7* and *SPL14* were significantly more highly expressed, while *SPL2*, *SPL5*, *SPL12*, *SPL15*, and *SPL16A* were prominently less expressed in Chuansang leaves, and the expression levels of *SPL* genes in Guisangyou 62 all decreased to varying degrees (Figure 7c,d and Appendix A). We inferred that the long period of artificial selection caused the different expression patterns of *SPL7* and *SPL14* genes between Chuansang and Guisangyou 62 under silkworm herbivory treatment. In addition, it was also found that a series of genes (*TT2L2*, *GL3*, *bHLH*, *TTG1*, *FNS*, *F3′H*, *DFR*, and *LAR*) involved in flavonoid biosynthesis were more highly expressed after herbivory treatment in Chuansang (Figure 8a,b). In mulberry, the TT2L1/bHLH/TTG1 or TT2L2/bHLH/TTG1 ternary complex regulates the transcription of genes associated with catechin synthesis [40]. Promoter analysis and dual-luciferase assay verified that SPL7 could promote the transcription of TT2L2 in Chuansang (Figure 9). However, similar experimental results were not found in Guisangyou 62 (Figure 8c). Moreover, it was also observed that miR156 had higher expression after herbivory treatment in Chuansang (Figure 7b) and had the same expression profile as miR156-targeted *SPL* genes (*SPL7* and *SPL14*). In conclusion, this study had found that *SPL* genes in Chuansang and Guisangyou 62 responded differently to silkworm herbivory, and verified that *SPL7*, independent of the miR156/SPL module, promoted the transcription of TT2L2 and further increased the expression levels of catechin synthesis genes (*F3′H*, *DFR*, and *LAR*) in response to silkworm herbivory in Chuansang.

## 4. Materials and Methods

### 4.1. Plant Materials and Growth Conditions

Guisangyou 62, Chuansang, and tobacco seeds (*Nicotiana tabacum* L.) were planted in sterilized soil and left at 4 °C for 2 d before transfer to a climate chamber at 25 °C under long-day conditions (16 h light/8 h dark), as were mulberry seedlings. Mature leaves from the juvenile phase of Guisangyou 62 (J-ML) and mature leaves from the mature phase of Guisangyou 62 (M-ML) were used to investigate the expression levels of *SPL* genes.

The 3-month-old Guisangyou 62 and Chuansang leaves were treated with second-stage silkworm herbivory (*Bombyx mori cv. Dazao*) for 1 h until obvious damage was caused to mulberry leaves.

### 4.2. Bioinformatics Analysis of SPL Genes in Mulberry

BLAST and HMMER searches of *MnSPLs* against the MorusDB (https://morus.swu.edu.cn/, accessed on 3 March 2020) were conducted using *AtSPL* as the query sequences. The SBP domain of SPLs was identified using the CD-search online tool (https://www.ncbi.nlm.nih.gov/Structure/cdd/wrpsb.cgi, accessed on 3 March 2020) (Appendix A).

Neighbor-joining phylogenetic trees of SPL proteins were constructed using MEGA5.1 with the best JTT + G model. Branching reliability was assessed by the bootstrap re-sampling method using 500 bootstrap replicates.

Chromosome locations of *SPL* genes in mulberry were determined by BLAST analysis of *SPLs* against the mulberry genome (https://www.ncbi.nlm.nih.gov/genome/?term=Morus+alba, accessed on 3 March 2020). The structures of *SPL* genes were predicted with the Gene Structure Display Server (http://gsds.cbi.pku.edu.cn/chinese.php, accessed on 3 March 2020). Sequence logos of SBP domains were generated by the Weblogo platform (http://weblogo.berkeley.edu/, accessed on 3 March 2020). Potential protein motifs were predicted using the MEME online tool (http://meme.sdsc.edu/meme/, accessed on 3 March 2020). Promoter analysis of gene was processed by the PlantPAN software (http://plantpan.itps.ncku.edu.tw/promoter.php, accessed on 3 March 2020), and the specific transcription factor binding motifs from *P. trichocarpa* and *A. thaliana* were selected as the query motifs.

### 4.3. Epression Analysis of miR156 and SPL Genes in Mulberry

The transcriptome data used in *SPL* gene expression profiling in five mulberry tissues was obtained from MorusDB (https://morus.swu.edu.cn/, accessed on 3 March 2020). The transcriptome data of *SPL* genes in mulberry leaves under silkworm herbivory are shown in Appendix A. The RPKM values of *SPL* genes were normalized through a min-max normalization algorithm (x∗ = (x − x_mean_)/(x_max_ − x_min_)), and analyzed using TBtools software [46].

Small RNAs (sequence lengths < 200 bp) and the total RNA of mulberry tissues were both extracted using the miRcute Plant miRNA Isolation Kit purchased from TIANGEN (DP504, Beijing, China), the OD260/280 values of all RNA samples were detected by NanoDrop2000 (Thermo Scientific, Waltham, USA), and agarose gel electrophoresis (AGE) was performed to verify the integrity of RNA samples (data not shown). Small RNA reverse transcriptions were performed using the miRNA First Strand cDNA Synthesis Kit (tailing reaction) (B532451, Sangon Biotech, Shanghai, China). The levels of mno-miR156 were quantified by the MicroRNA qPCR Kit (SYBR Green method) (B532461, Sangon Biotech, Shanghai, China). For reference genes, mno-miR166b and MnU6 were selected to calibrate data in small RNA RT-qPCR. The relative expression of miR156 was defined using the 2^−[Ct(target gene) − Ct(control gene)]^ algorithm. The cDNA of total RNA was synthesized according to the instructions of Primer Script RT reagent kit (RR047A, Takara, Japan). For the RT-qPCR, a reaction was performed according to the manufacturer’s instructions for the SYBR Premix Ex Taq II (RR820A, Takara, Japan) and processed using a Step One Plus Real-Time PCR System (Applied Biosystems, Singapore, Singapore). The mulberry *ribosomal protein L15* (*RPL15*, *Morus024083*) gene was used as a control for expression normalization, and the relative expression of genes was defined using the 2^−[Ct(target gene) − Ct(control gene)]^ algorithm. Gene-specific primers used for RT-qPCR are listed in Appendix A.

### 4.4. Dual-Luciferase Repoter Assay

*MnSPL* genes were cloned into the GreenII 62-SK vector. A 2000-bp promoter sequence of TT2L2 predicted by the Promoter 2.0 Prediction Server (http://www.cbs.dtu.dk/services/Promoter/, accessed on 31 May 2020) was amplified and cloned into the pGreenII 0800-LUC vector. Agrobacterium-mediated co-transformation of the pGreenII 0800-LUC and GreenII 62-SK vectors into tobacco leaves was performed as described previously [40]. After infiltration for 2 days, the ratio of LUC/REN activity was measured using the Dual-Luciferase Reporter Gene Assay Kit (11402ES80, Yeasen, Shanghai, China) on a configurable multi-mode microplate reader (Synergy™ H1, BioTek, Beijing, China). All the primers and probes used in this work are listed in Appendix A.

## 5. Conclusions

In this study, a total of 15 full-length SPLs were identified in the mulberry genome. The evolutionary conservation and functional diversity of mulberry *SPL* genes were inferred through a comprehensive analysis of chromosomal locations, phylogenetic relationships, gene structures, conserved motifs, and spatial and temporal expression profiles. In addition, we found that *SPL* genes in cultivated mulberry (*M. atropurpurea cv. Guisangyou 62*) and wild mulberry (*M. notabilis*) were differentially expressed after silkworm herbivory and verified that the SPL7/TT2L2 module increased the expression levels of catechin synthesis genes (F3′H, DFR, and LAR) to response to silkworm herbivory in Chuansang. Our work provides useful information to elucidate the functions of SPLs in mulberry.

## Figures and Tables

**Figure 1 ijms-23-01141-f001:**
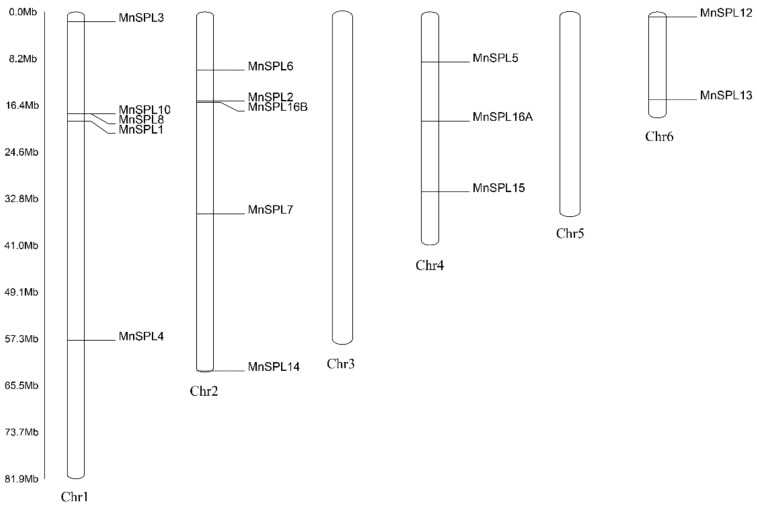
Distribution of *SPL* genes in the mulberry genome. The scale is presented on the left.

**Figure 2 ijms-23-01141-f002:**
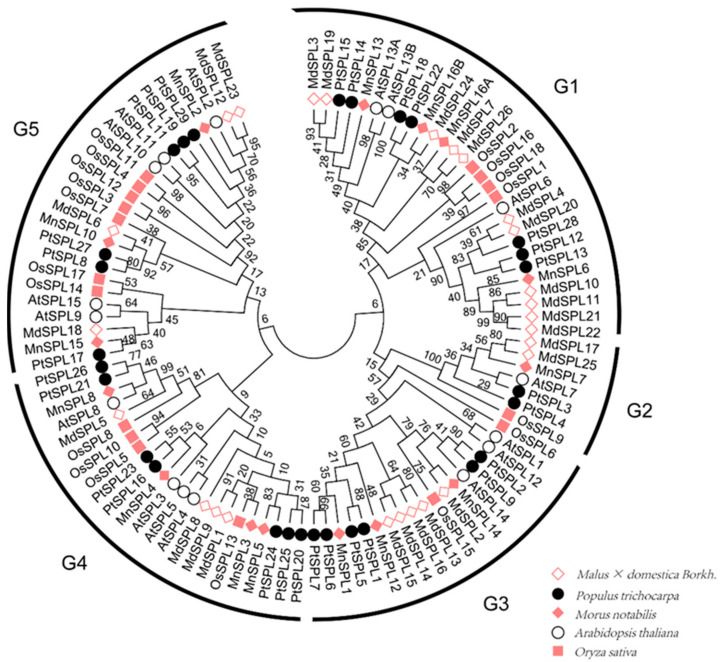
Phylogenetic tree of the SPL family based on the amino acid sequences of SBP domains. Posterior probability values of nodes are shown near the nodes. Different shapes represent different plant species (Md: *Malus × domestica Borkh*; Pt: *P. trichocarpa*; Mn: *M. notabilis*; At: *A. thaliana*; Os: *O. sativa*).

**Figure 3 ijms-23-01141-f003:**
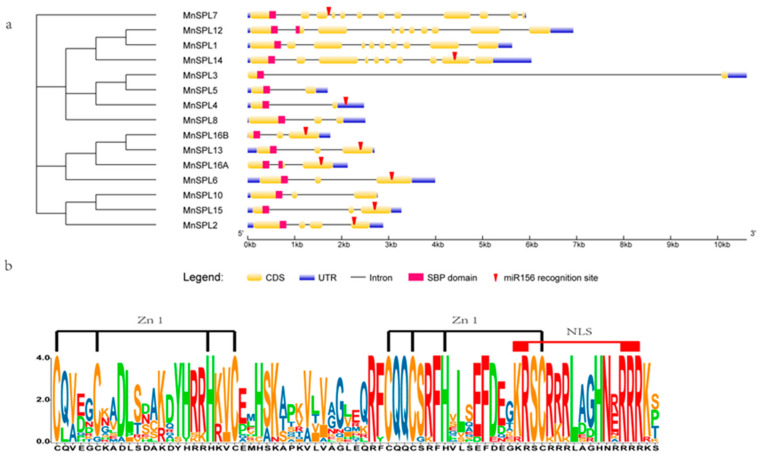
The structures of the full-length *SPL* genes in mulberry. (**a**) Gene structure diagram of *MnSPL* genes. (**b**) Sequence logo of the SBP domain of MnSPLs. The height of the letter (amino acid) at each position represents the degree of conservation.

**Figure 4 ijms-23-01141-f004:**
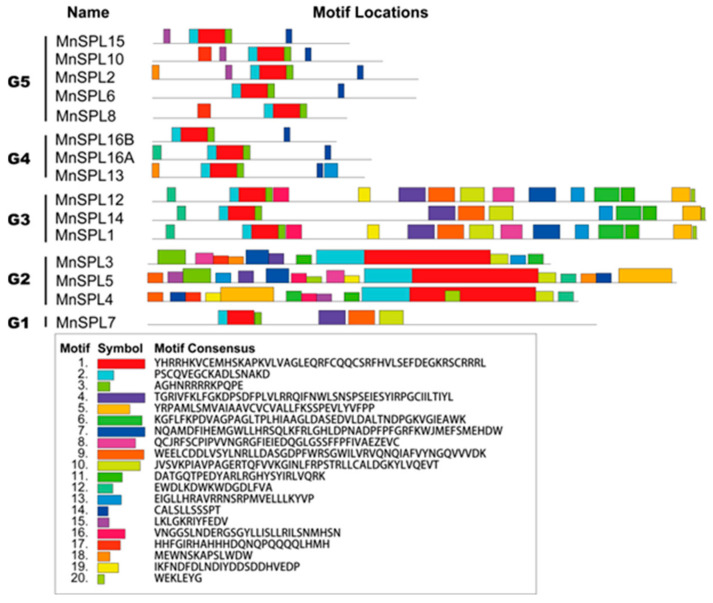
Distribution of putative conserved motifs in MnSPLs. The legend of the conserved motifs is shown below the graphic. Numbers 1–20 represent motifs 1–20, respectively. Box size indicates the length of motifs.

**Figure 5 ijms-23-01141-f005:**
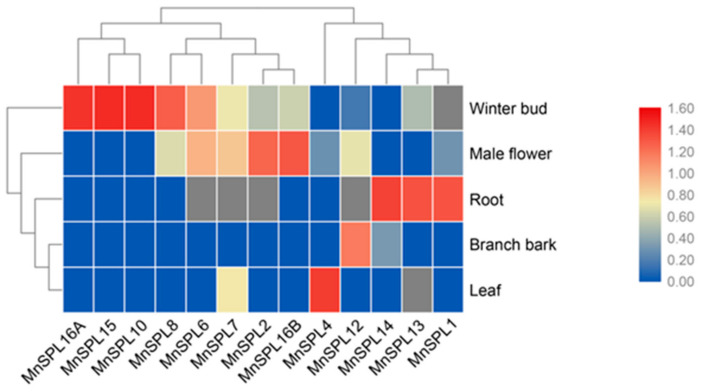
Tissue-specific expression patterns of *MnSPL* genes. Transcriptome data were obtained from the Morus Genome Database (MorusDB) (https://morus.swu.edu.cn/, accessed on 3 March 2020). Blue indicates that the expression levels of *SPL* genes are low, while red indicates that the levels are high.

**Figure 6 ijms-23-01141-f006:**
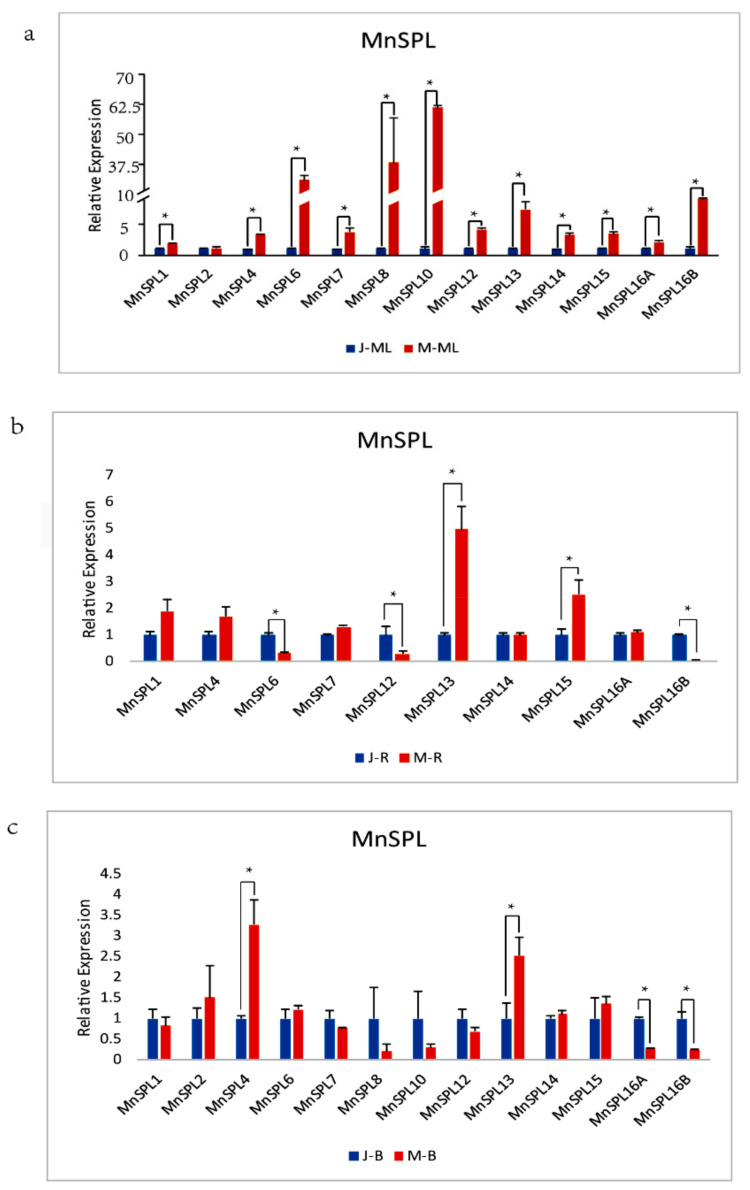
The expression profiles of *MnSPL* genes in the juvenile and mature phases of 3 mulberry tissues. (**a**) Leaves, J-ML: mature leaves at the juvenile phase of mulberry, M-ML: mature leaves at the mature phase of mulberry. (**b**) Roots, J-R: roots at the juvenile phase of mulberry, M-R: roots at the mature phase of mulberry. (**c**) Bark. J-B: bark at the juvenile phase of mulberry, M-B: bark at the mature phase of mulberry. Values represent the mean ± SD of 3 biological replicates and were statistically analyzed (independent samples *t*-test): * *p* < 0.05.

**Figure 7 ijms-23-01141-f007:**
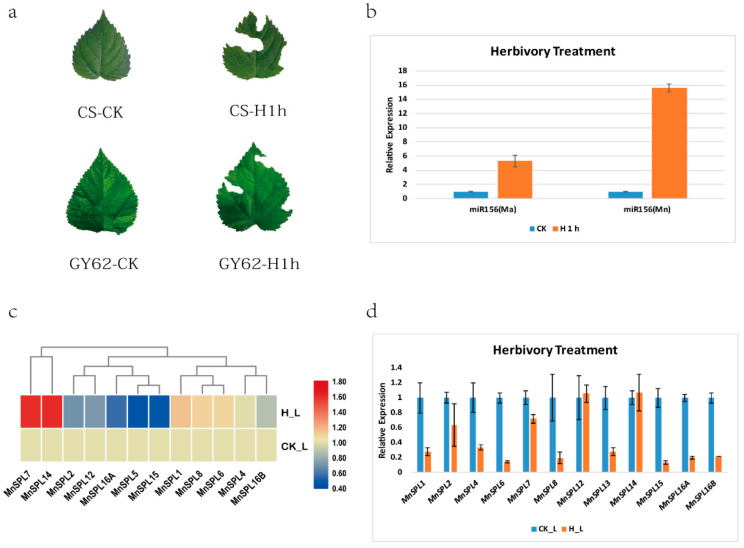
The expression pattern analysis of *SPL* genes in mulberry leaves under herbivory treatment. (**a**) Mulberry leaves with 1 h of herbivory treatment. CS-CK: the control group of Chuansang. CS-H1h: Chuansang leaves after herbivory treatment for 1 h. GY62-CK: the control group of Guisangyou 62. GY62-H1h: Guisangyou 62 leaves after herbivory for 1 h. (**b**) The expression profiles of mulberry miR156 after herbivory treatment. Mn: *M. notabilis*. Ma: *M. alba*. (**c**) The expression profiles of *SPL* genes in mulberry (*M. notabilis*) leaves after herbivory treatment. (**d**) The expression profiles of *SPL* genes in mulberry (*M. alba*) leaves after herbivory treatment. H_L: Leaves under herbivory treatment. CK_L: Control mulberry leaves with no herbivory treatment.

**Figure 8 ijms-23-01141-f008:**
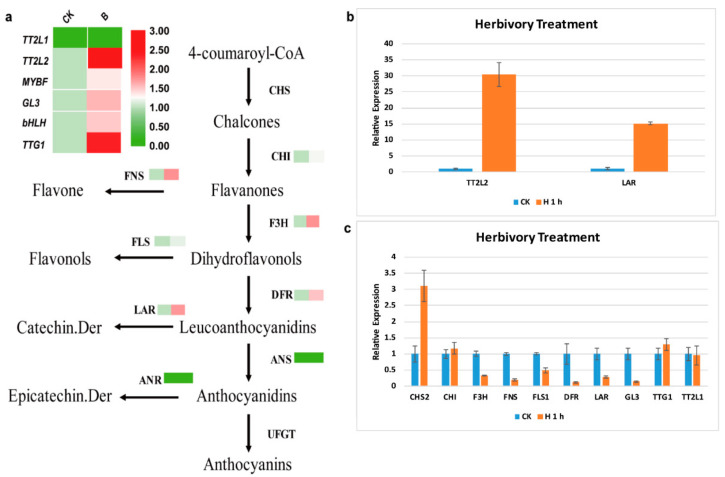
The expression profiles of genes involved in the mulberry flavonoid synthesis pathway after silkworm herbivory treatment. (**a**) The expression profiles of genes involved in the Chuansang flavonoid synthesis pathway. (**b**) The expression trend of *TT2L2* and *LAR* in Chuansang. (**c**) The expression pattern of genes involved in the Guisangyou 62 flavonoid synthesis pathway.

**Figure 9 ijms-23-01141-f009:**
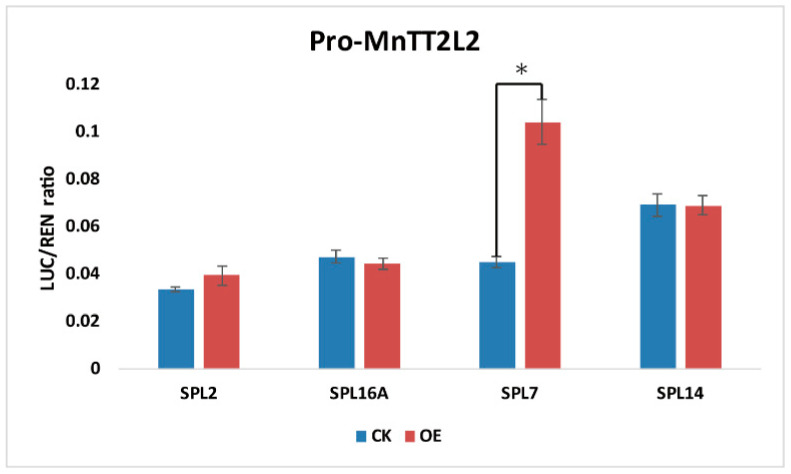
Dual-luciferase assays identified the interaction between *SPL* genes and the promoter of *MnTT2L2*. CK: Control group. OE: overexpressed mulberry *SPL* genes. Values represent the mean ± SD of 3 biological replicates and were statistically analyzed (independent samples *t*-test): * *p* < 0.05.

**Table 1 ijms-23-01141-t001:** Gene ID and gene structures of *SPLs* in mulberry. The gene ID of MnSPLs were obtained from the Morus Genome Database (MorusDB) (https://morus.swu.edu.cn/, accessed on 3 March 2020).

Gene Name	Gene ID	mRNA Length	CDS Length	Exon Number	Strand	miR156 Target Site
MnSPL1	Morus013868	5282	3081	10	-	/
MnSPL2	Morus015493	2478	1503	4	-	GUGCUCUCUCUCUUCUGUCAA
MnSPL3	Morus009607	10,219	480	2	+	/
MnSPL4	Morus014488	1855	513	2	-	UUGCUCUCUCUCUUCUGUCAA
MnSPL5	Morus010322	1381	630	2	-	/
MnSPL6	Morus026457	3240	1491	3	-	GUGCUCUCUCUCUUCUGUCAU
MnSPL7	Morus011281	5842	2535	12	+	GAUGUCUCUUUCCUCUGUCAG
MnSPL8	Morus021788	2020	1095	3	-	/
MnSPL10	Morus021787	2565	1302	3	+	/
MnSPL12	Morus025152	6389	3072	10	-	/
MnSPL13	Morus010123	2469	1200	3	-	GUGCUCUCUAUCUUCUGUCAU
MnSPL14	Morus024784	5168	3129	10	-	UUGCUCAC-GUUUUCUGUUGA
MnSPL15	Morus018032	2947	1110	3	+	GUGCUCUCUCUCUUCUGUCAA
MnSPL16A	Morus010792	1816	1239	3	-	GUGCUCUCUAUCUUCUGUCAA
MnSPL16B	Morus017456	1515	1041	3	-	GUGCUCUCUCUCUUCUGUCAU

## Data Availability

Publicly available datasets were analyzed in this study. This data can be found here: https://morus.swu.edu.cn/, accessed on 3 March 2020, https://www.arabidopsis.org/, accessed on 3 March 2020, and http://www.phytozome.net/poplar.php, accessed on 3 March 2020, Refs. [45,47].

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
