# Peer review of "The Mulberry SPL Gene Family and the Response of MnSPL7 to Silkworm Herbivory through Activating the Transcription of MnTT2L2 in the Catechin Biosynthesis Pathway"

_ijms, 2022, doi:10.3390/ijms23031141_

Round 1

Reviewer 1 Report

In the current manuscript, Li and colleagues summarized the SPL gene family through the whole genome level from mulberry. Gene expression characteristics demonstrate the tissue-specific of these different SPL genes in mulberry. Additionally, the author focused on two SPL genes, SPL7 and SPL14, looks like these two genes respond to the silkworm herbivory regulated downstream genes through transcription factor TT2L2.

However, the data revealed in the manuscript is very hard to support this paper to be published in this journal.  The following is the main questions that reader may be concerned about the manuscript:

  1. English description should be improved carefully of the whole manuscript
  2. In the 1st section of the results, the author paid more attention to emphasizing the gene family of the SPL gene family, but in the latter parts, the data of SPL7 and SPL14 is very weak. The logic of the whole manuscript makes the reader confused that what kind of a story that the author wants to express. If just for searching for the SPL gene family, more detail of the SPL gene family and evolution analysis should be performed in the manuscripts.
  3. If want to focus on the internal relationship between the SPL4 and SPL14 to the flavonoid biosynthesis, and the transcription factors, just gene expression is far away to make a conclusion.
  4. What’s the relationship between the Micro156 and SPL, the author did not explain this very well.
  5. In the flavonoid pathway, why the gene expression data of Guisang and Chuansang is various a lot?
  6. The promoter and effectors analysis data are very important, should be shown.
  7. More detail of protein-promoter interaction results should be demonstrated to show the reliable of this work.

Author Response

Dear Editor,

We are resubmitting the revised manuscript (manuscript ID: ijms-1524708) entitled " Mulberry SPL gene family and the response of MnSPL7 to silkworm herbivory through activating the transcription of MnTT2L2 in catechins biosynthesis pathway" by Li et al again for publication in International Journal of Molecular Sciences. In this revised version, we have addressed two reviewers’ comments. We have thoroughly checked the language. In order for you to know which parts have been changed, we highlighted all of these rewritten sentences and words in gray. Now we are listing the detailed point-to-point response as follows.

We deeply appreciate your consideration of our manuscript and the helpful suggestions offered by reviewers. We have studied their comments carefully and made corrections. We hope you now find that this revised manuscript is suitable for publication.

Thank you in advance for your help.

Sincerely,

Ningjia He

Professor

State Key Laboratory of Silkworm Genome Biology, Southwest University, Beibei, Chongqing 400715, China

Tel: +86-23-6825-0797

Fax: +86-23-6825-1128

Response to reviewer 1 comments

Point 1: English description should be improved carefully of the whole manuscript.

Response 1: According to your suggestion, we have thoroughly checked the whole manuscript and highlighted all of the rewritten sentences and words in gray.

Point 2: In the 1st section of the results, the author paid more attention to emphasizing the gene family of the SPL gene family, but in the latter parts, the data of SPL7 and SPL14 is very weak. The logic of the whole manuscript makes the reader confused that what kind of a story that the author wants to express. If just for searching for the SPL gene family, more detail of the SPL gene family and evolution analysis should be performed in the manuscripts. If want to focus on the internal relationship between the SPL4 and SPL14 to the flavonoid biosynthesis, and the transcription factors, just gene expression is far away to make a conclusion

Response 2: miR156/SPL model play an important role in plant development and stress responses. In our previous study, we had carried out a systematic study about the role of miR156/SPL model in mulberry development. After that, we pay more attention to the role of mulberry SPLs under silkworm herbivory. We then focus on the SPL genes with significant different expression profile under silkworm herbivory treatment. The aim of the present study is to provide information of mulberry SPLs through comprehensive analyses of gene structures, phylogenetic relationships, chromosomal locations, conserved motifs, and expression patterns. Such knowledge will facilitate future functional identification of SPL genes from closely related plant species.

Point 3: What’s the relationship between the Micro156 and SPL, the author did not explain this very well.

Response 3: As you suggested, the descriptions of relationship between the miR156 and SPL genes have been added and the rewrote sentences have been highlighted in gray at lines 50 and 51.

Point 4: In the flavonoid pathway, why the gene expression data of Guisang and Chuansang is various a lot?

Response 4: The flavonoid pathway is mainly regulated by MYB–bHLH–WD40 (MBW) transcription complexes. TT2L1 and TT2L2 work with bHLH3 or GL3 and form a MYB-bHLH-WD40 (MBW) complex with TTG1 to regulate proanthocyanidin (PA) synthesis through up-regulate the expression of PA synthesis structural genes including flavanone 3-hydroxylase (F3H), dihydroflavonol 4-reductase (DFR), and leucoanthocyanidin reductase (LAR). A variety of flavonoid synthesis structural genes will be regulated. Considering the different genetic background of Guisang and Chuansang, the gene expression data of these two mulberry resources should be various a lot.

Point 5: The promoter and effectors analysis data are very important, should be shown.

Response 5: The result of promoter analysis of MnTT2L2 has been added and shown in a new supplementary table, Table S4.

Point 6: More detail of protein-promoter interaction results should be demonstrated to show the reliable of this work.

Response 6: Thank you for your suggestion. We have attempted to confirm the interaction between SPLs and the promoter of MnTT2L2 by Dual-luciferase reporter system and Yeast one-hybrid (Y1H) assay. However, the yeast cells containing pBait-AbAi-MnTT2L2-promoter vector could not be effectively suppressed by high-concentration Aureobasidin A (1000 ng/ml), indicating the promoter sequence of MnTT2L2 performed self-activation in the yeast cells. The interaction data between SPLs and the promoter of MnTT2L2 had been obtained only by Dual-luciferase assay.

Reviewer 2 Report

The article presented is very well presented. The methods are adequately described and the discussion is well prepared. The phylogenetic tree of SPL family is very interesting and I think it will be considered by other authors in the future. The Reference section contains relevant, new paperes. In my opinion, the article may be published in present form, without any changes.  

Author Response

Dear Editor,

We are resubmitting the revised manuscript (manuscript ID: ijms-1524708) entitled " Mulberry SPL gene family and the response of MnSPL7 to silkworm herbivory through activating the transcription of MnTT2L2 in catechins biosynthesis pathway" by Li et al again for publication in International Journal of Molecular Sciences. In this revised version, we have addressed two reviewers’ comments. We have thoroughly checked the language. In order for you to know which parts have been changed, we highlighted all of these rewritten sentences and words in gray. Now we are listing the detailed point-to-point response as follows.

We deeply appreciate your consideration of our manuscript and the helpful suggestions offered by reviewers. We have studied their comments carefully and made corrections. We hope you now find that this revised manuscript is suitable for publication.

Thank you in advance for your help.

Sincerely,

Ningjia He

Professor

State Key Laboratory of Silkworm Genome Biology, Southwest University, Beibei, Chongqing 400715, China

Tel: +86-23-6825-0797

Fax: +86-23-6825-1128

Response to reviewer 2 comments

Point 1: The article presented is very well presented. The methods are adequately described and the discussion is well prepared. The phylogenetic tree of SPL family is very interesting and I think it will be considered by other authors in the future. The Reference section contains relevant, new papers. In my opinion, the article may be published in present form, without any changes.

Response: We appreciate the comments of reviewer 2.

Reviewer 3 Report

Mulberry SPL gene family and the response of MnSPL7 to silkworm herbivory through activating the transcription of MnTT2L2 in catechins biosynthesis pathway

Authors in this study have identified the Squamosa promoter binding protein like genes in mulberry genome to study the role of these genes in development and stress response of mulberry. A phylogenetic analysis has been conducted on the 15 SPL genes identified from the genome and an expression analysis in different tissues and developmental stages was performed. Additionally, the expression patterns of SPL genes in the pre-existing transcriptome of mulberry leaves fed upon by silkworm was identified.

Overall the authors made an effort to use the pre-existing transcriptome database to identify the expression profiles of the SPL genes. However, the manuscript can be improved in writing especially in highlighting the connection between the miR56 and SPL genes better. The information provided in the introduction is not sufficient to understand the relationship of the SPL and catechin biosynthesis pathway. The figure quality is really poor. Citation for the MorusDB is missing. The manuscript needs significant improvement in writing and connecting the concepts discussed in the submitted manuscript.

Author Response

Dear Editor,

We are resubmitting the revised manuscript (manuscript ID: ijms-1524708) entitled " Mulberry SPL gene family and the response of MnSPL7 to silkworm herbivory through activating the transcription of MnTT2L2 in catechins biosynthesis pathway" by Li et al again for publication in International Journal of Molecular Sciences. In this revised version, we have addressed two reviewers’ comments. We have thoroughly checked the language. In order for you to know which parts have been changed, we highlighted all of these rewritten sentences and words in gray. Now we are listing the detailed point-to-point response as follows.

We deeply appreciate your consideration of our manuscript and the helpful suggestions offered by reviewers. We have studied their comments carefully and made corrections. We hope you now find that this revised manuscript is suitable for publication.

Thank you in advance for your help.

Sincerely,

Ningjia He

Professor

State Key Laboratory of Silkworm Genome Biology, Southwest University, Beibei, Chongqing 400715, China

Tel: +86-23-6825-0797

Fax: +86-23-6825-1128

Response to reviewer 3 comments

Point 1: The manuscript can be improved in writing especially in highlighting the connection between the miR56 and SPL genes better.

Response 1: According to your suggestion, the detailed introduction between the miR156 and SPL genes has been added and the rewrote sentences were highlighted in gray at lines 50 and 51.

Point 2: The information provided in the introduction is not sufficient to understand the relationship of the SPL and catechin biosynthesis pathway.

Response 2: As you pointed out, the relationship of TT2L2 (the downstream genes of SPL7, which was identified in our work) and catechin biosynthesis pathway have been provided at lines 72 and 73.

Point 3: The figure quality is really poor

Response 3: As your suggestion, we have improved the resolution ratio of pictures of Fig. 6-9 in this version.

Point 4: Citation for the MorusDB is missing

Response 4: The citation of MorusDB has been added at lines 473 and 474.

Point 5: The manuscript needs significant improvement in writing and connecting the concepts discussed in the submitted manuscript.

Response 5: According to your suggestion, we have thoroughly checked this paper and highlighted all of these rewritten sentences and words in gray.

Round 2

Reviewer 1 Report

After a round of revision, the expression of the current manuscript was improved a lot, additionally, the answers related to the questions about this manuscript are basically reliable, could be considered to be published in this journal.

Reviewer 3 Report

The authors have incorporated the suggestions and the manuscript has improved in quality. There are minor typos which need to be corrected. I agree for the manuscript to be published in IJMS.